# Association Between Shift Work and Vitamin D Levels in Brazilian Female Workers

**DOI:** 10.3390/nu17071201

**Published:** 2025-03-29

**Authors:** Ingrid Stähler Kohl, Anderson Garcez, Janaína Cristina da Silva, Harrison Canabarro de Arruda, Raquel Canuto, Vera Maria Vieira Paniz, Maria Teresa Anselmo Olinto

**Affiliations:** 1Pos-Graduate Program in Medical Sciences: Endocrinology, Faculty of Medicine, Federal University of Rio Grande do Sul State, UFRGS, Porto Alegre CEP 90035-002, RS, Brazil; ingridkohl.nutri@gmail.com (I.S.K.); adsgarcez@gmail.com (A.G.); 2Post-Graduate Program in Food, Nutrition and Health, Faculty of Medicine, Federal University of Rio Grande do Sul, UFRGS, Porto Alegre CEP 90035-002, RS, Brazil; 3Post-Graduate Program in Collective Health, University of Vale do Rio dos Sinos, Unisinos, São Leopoldo CEP 93022-750, RS, Brazil; 4Hospital de Clínicas de Porto Alegre, Federal University of Rio Grande do Sul, UFRGS, Porto Alegre CEP 90035-903, RS, Brazil

**Keywords:** shift work, night work, vitamin D, occupational health, women

## Abstract

**Objectives**: Vitamin D plays a crucial role in maintaining bone and muscle health and is associated with various health conditions. Risk factors for reduced vitamin D levels include occupation. The aim of this study was to explore the association between shift work and vitamin D levels among female workers. **Methods**: This cross-sectional study was conducted among 304 women from an industrial group located in southern Brazil. Vitamin D deficiency was defined as serum 25(OH)D levels < 20 ng/mL, and deficiency/insufficiency was defined as serum levels < 30 ng/mL. Work shift data were collected through interviews using the start and end times of work shifts, classified as “day shift” (6:00 AM ≤ hh:hh < 10:00 PM) and “night shift” (10:00 PM ≤ hh:hh < 6:00 AM), respectively. The association between vitamin D deficiency and shift work was expressed as prevalence ratio (PR), using Poisson regression adjusted for confounding variables. **Results**: The prevalence of vitamin D deficiency was 36.5% (95% CI: 31.1–41.9), while the prevalence of insufficiency was 75.7% (95% CI: 70.8–80.5). After adjustment, a significant association was found, with 65% and 17% higher probabilities of having vitamin D deficiency (PR = 1.65; 95% CI: 1.23–2.22; *p* = 0.001) and vitamin D deficiency/insufficiency (PR = 1.17; 95% CI: 1.02–1.35; *p* = 0.030) among nightshift workers compared with dayshift workers. **Conclusions**: The findings of this study indicate a high prevalence of hypovitaminosis D among Brazilian female fixed-shift workers as well as a higher probability of vitamin D deficiency and deficiency/insufficiency among nightshift workers compared with dayshift workers.

## 1. Introduction

Vitamin D is a fat-soluble vitamin recognized as a prohormone regulated by metabolic feedback cycles [1]. It plays a fundamental role in maintaining bone and muscle health, being crucial for calcium and phosphorus absorption [2]. Moreover, vitamin D strengthens the immune system and is associated with the regulation of various metabolic functions, such as glycemic control and blood pressure regulation [3,4]. Serum 25-hydroxyvitamin D (25[OH]D) levels have been used as a measure of adequacy for vitamin D in the body, as this form reflects both dietary intake and the amount produced through cutaneous synthesis via sun exposure [2]. However, the ideal blood concentration levels of 25(OH)D for adults remain a topic of debate, given that vitamin D is associated with various health outcomes [5].

The global prevalence of vitamin D deficiency (<20 ng/mL) has been estimated at 47.9%, while the prevalence of individuals with levels below 30 ng/mL has reached 76.6% [6]. In Brazil, the prevalence of deficiency and serum levels below 30 ng/mL are approximately 30% and 45%, respectively, with higher frequencies observed in the southern and southeastern regions of the country [7]. Overall, higher frequencies are observed in women than in men—approximately 1.3 times higher [6,8,9]. Among women, vitamin D deficiency is associated with various adverse pregnancy outcomes, such as an increased risk of preeclampsia, gestational diabetes mellitus, and postpartum depression [10,11,12].

The primary source of vitamin D is exposure to natural sunlight [1,13]. Nevertheless, high prevalence rates of vitamin D deficiency are observed even in countries with abundant sunlight exposure [6]. Risk factors associated with vitamin D deficiency include latitude, season, race, genetic conditions, and lifestyle factors, such as duration of sun exposure, type of clothing, use of sunscreen, and dietary habits, among others [5]. Systematic reviews have shown that occupation is also an important factor contributing to reduced vitamin D levels. In general, it has been observed that indoor workers are at a higher risk of vitamin D deficiency than outdoor workers [14,15,16].

Shift work is organized according to a specific pattern, which may be rotating or fixed, commonly practiced in sectors with continuous operations and often including night shifts [17]. In this context, shift work also appears to be an occupational factor associated with serum vitamin D levels. Significantly lower serum vitamin D levels have been observed in shift workers, as they have fewer opportunities for sun exposure than nonshift workers [18], with nightshift workers presenting a higher prevalence of vitamin D deficiency compared with dayshift workers [19,20]. Thus, the aim of this study was to explore the association between shift work and vitamin D levels among female workers. Our study hypothesis is that nightshift workers have a higher prevalence of vitamin D deficiency.

## 2. Materials and Methods

### 2.1. Study Design and Population

This cross-sectional study was conducted among a sample of female workers from a large industrial group engaged in the manufacturing of various segments of plastic products and household utensils, headquartered in the metropolitan region of Porto Alegre, Rio Grande do Sul, Brazil. This study is part of a larger research project entitled “Health Conditions of Shift-Working Women: Longitudinal Occupational Health Study (ELO Saúde)”. This study was previously submitted to and approved by the Ethics Committee of University do Vale do Rio dos Sinos: CAAE no. 53762521.7.0000.5344/Approval no. 5681627. All participating workers signed an informed consent form adhering to ethical procedures (informed consent, confidentiality, and anonymity) in accordance with the Helsinki Declaration on research ethics involving human subjects.

### 2.2. Sample and Sampling

All female workers aged 18 years or older from three factories that specialize in the manufacturing of household plastic products were considered eligible. This study included workers from both the production and administrative sectors. The exclusion criteria established were pregnancy, temporary absence from work, and employment at the company for less than three months. Of the 546 eligible female workers, 452 were interviewed after accounting for losses and refusals. All interviewed participants underwent blood sample collection, resulting in laboratory data from 304 workers, including 232 production sector workers engaged in fixed shifts with a six-day workweek schedule, and 72 administrative sector workers.

### 2.3. Data Collection and Instruments

Data were collected between August 2022 and March 2023. The laboratory tests were conducted during the same period, covering the winter, spring, and summer seasons in Brazil. A standardized, precoded, and pretested questionnaire was applied through in-person interviews at the participants’ workplaces. A pilot study was conducted to test the instruments and train the interviewers. To ensure data quality control, 10% of the interviews were repeated through telephone contact using a simplified questionnaire with questions containing responses not subject to short-term changes. Blood samples were collected by trained professionals at a designated workplace location or at the participant’s home, and laboratory tests were conducted by a specialized company. For biological material collection, participants were instructed to fast for at least 8 h and no more than 12 h, refrain from consuming alcoholic beverages in the 72 h prior, avoid caffeine intake, and abstain from vigorous exercise for 24 h before collection. Data coding was performed by research supervisors.

### 2.4. Outcomes: Vitamin D Deficiency/Insufficiency

Serum levels of 25(OH)D were obtained and measured using serum samples (minimum volume of 2.0 mL). The samples were centrifuged at 5000 rpm for 10 min at 18 °C and analyzed using the Chemiluminescent Microparticle Immunoassay (CMIA) method, with the results expressed in ng/mL. Vitamin D deficiency was defined as a serum level < 20 ng/mL (cutoff point according to the Institute of Medicine) [1], while vitamin D deficiency/insufficiency was defined as a serum level < 30 ng/mL (cutoff point according to the Endocrine Society) [13].

### 2.5. Main Exposure: Shift Work

All study participants were interviewed during their work shifts. The permanent production line shift workers, who work a total of 44 h per week, follow a schedule of six consecutive fixed shifts (Monday through Saturday), followed by one day off, and are distributed across the following fixed shifts: morning (06:00 AM ≤ hh:hh < 02:00 PM), afternoon (02:00 PM ≤ hh:hh < 10:00 PM), and night (10:00 PM ≤ hh:hh < 06:00 AM). Administrative sector workers with work schedules falling within the daytime hours (07:00 AM ≤ hh:hh < 07:00 PM) were also included. Data on the start and end times of the participants’ work shifts were collected, and each participant was subsequently classified as a “day shift” worker (06:00 AM ≤ hh:hh < 10:00 PM) or a “night shift” worker (10:00 PM ≤ hh:hh < 06:00 AM).

### 2.6. Explanatory Variables (Covariates)

Data on demographic, socioeconomic, behavioral, and occupational characteristics were collected to characterize the population sample and to control for potential confounding factors in a multivariable analysis. The following demographic and socioeconomic characteristics were considered: age, reported in complete years at the time of the interview and categorized into age groups (18–30, 31–40, ≥41 years); skin color/race, self-reported by the interviewee and categorized as white or other (black, mixed-race, indigenous, or Asian); marital status, reported by the interviewee and classified as “without a partner” (single/separated/divorced/widowed) or “with a partner” (married/in a relationship); education level, reported in completed years of study and categorized as ≤8, 9–11, or ≥12 years; per capita household income, calculated by summing the reported income of each family member in the last month and dividing it by the number of household residents, categorized in terms of minimum wages (<1, ≥1–≤2, >2 minimum wages; based on the 2022 national minimum wage: BRL 1212.00); and head of household, reported as having (yes) or not having (no) responsibility for household finances among family members. The behavioral characteristics investigated included leisure-time physical activity, assessed on the report of practicing any physical activity for leisure, sports, or exercise in the past week, excluding commuting—the response was categorized as “no” or “yes”; current smoking status, reported and categorized as non-smoker, former smoker, or smoker; alcohol consumption—the response was classified as “no” (no consumption or consumption less than once per week) or “yes” (regular consumption of at least once per week in the past year); fruit/vegetable consumption— the response was classified as “no” (no consumption) or “yes” (consumption of fruits and/or vegetables in the week prior to the interview); usual sleep duration, assessed on total hours of sleep per day, according to the reported usual sleep and wake times during the week, and categorized as “>6 h per day” or “≤6 h per day”; number of pregnancies, reported and categorized as: none, 1 pregnancy, or 2 or more pregnancies; and nutritional status, assessed by Body Mass Index (BMI) using the average of two measurements of weight and height, and classified according to the WHO recommendation for adults [21]. Occupational variables investigated included length of employment, reported in months and categorized as ≤12, 13–36, or >36 months; and job role, referring to the position held at the time of the interview and categorized as factory (production) or administrative work.

### 2.7. Statistical Analyses

Data entry was performed using EpiData software version 3.1 (Centers for Disease Control and Prevention, Atlanta, GA, USA) following the double-entry procedure, comparison of data entries, and consistency analysis between them. Descriptive statistics were used to present the distribution of sample characteristics and vitamin D deficiency and deficiency/insufficiency outcomes. Numerical variables were described using means and standard deviations and/or medians and interquartile ranges (IQR), while categorical variables were presented as absolute (n) and relative (%) frequencies. Wilcoxon, Mann–Whitney, and Kruskal–Wallis tests were used to compare non-parametric data (distribution/medians of vitamin D levels according to sample characteristics). Pearson’s chi-square test was applied to compare the distribution (prevalence) of vitamin D deficiency and deficiency/insufficiency according to the sample characteristics.

To investigate the association between shift work (main exposure) and vitamin D deficiency and deficiency/insufficiency (study outcomes), prevalence ratios (PRs) and their respective 95% confidence intervals (95% CI) were obtained using Poisson regression with robust variance [22]. Three analytical models were explored, using the following hierarchical model for inclusion of variables in the adjusted (multivariate) analysis [23]: Model I—unadjusted analysis; Model II—analysis adjusted for demographic, socioeconomic, and behavioral characteristics; and Model III—analysis adjusted for Model II + occupational characteristics. Only variables with a significance level of 20% or less (*p* ≤ 0.20) in the bivariate analysis were included in the multivariate analysis (Models II and III). All analyses were performed using Stata software version 14.0 (StataCorp LP, College Station, TX, USA), considering associations with a *p*-value less than 5% (*p* < 0.05) as statistically significant.

## 3. Results

A total of 304 female workers aged 18–64 years (mean age: 35.4 ± 10.1 years) were included in the final analysis of this study. Table 1 presents the general characteristics of the sample. The majority were women aged 40 years or younger (69.8%), were white (69.7%), had no partner (51.6%), had completed 9 to 11 years of education (53.3%), had a per capita household income of 1 to 2 minimum wages (41.9%), and were not household heads (65.1%). Regarding behavioral characteristics, the majority never smoked (76.3%), did not consume alcohol regularly (68.8%), and consumed adequate fruit and vegetables (77.0%). However, only 29.6% of the workers engaged in leisure-time physical activity, and 46.7% had six or fewer hours of sleep per day. Regarding occupational characteristics, the majority had been employed for >3 years (49.4%), worked in the factory/production sector (76.3%), and had day shifts (86.8%). When comparing the distribution of characteristics according to work shifts, we observed differences between dayshift and nightshift workers in education level, leisure-time physical activity, usual sleep duration, nutritional status, and job role. All the nightshift workers were production workers. The vast majority (77.5%) had 9–11 years of education, while only 17.5% had 12 or more years of education. Additionally, 82.5% did not engage in leisure-time physical activity, 62.2% slept 6 h or less per day, and 52.5% were obese (Table 1).

Regarding laboratory measures, the mean and median serum vitamin D levels in the total sample were 24.1 ± 9.8 ng/mL and 22.9 ng/mL (IQR 17.4–29.1), respectively. Participants in the “other” race/skin color category had lower serum vitamin D levels (20.1 ng/mL; IQR: 14.7–27.9) compared with those who were white (23.6 ng/mL; IQR: 18.4–30.8; *p* = 0.006). The median serum level was significantly lower among nightshift workers (18.2 ng/mL; IQR: 13.7–24.3) than among dayshift workers (23.5 ng/mL; IQR: 18.2–30.8; *p* < 0.001) (Table 1).

As shown in Table 2, the prevalence rates of vitamin D deficiency (<20 ng/mL) and deficiency/insufficiency (<30 ng/mL) were 36.5% (95% CI: 31.1–41.9) and 75.7% (95% CI: 70.8–80.5), respectively. Table 3 presents the PRs for the association between shift work and vitamin D deficiency (<20 ng/mL) and deficiency/insufficiency (<30 ng/mL). In the unadjusted analysis (Model I), considering vitamin D deficiency (<20 ng/mL), nightshift workers showed an 82% higher probability of having vitamin D deficiency (PR = 1.82; 95% CI: 1.34–2.47; *p* < 0.001) than dayshift workers. After adjusting for confounding factors (Model III), the association remained significant, with a 65% higher probability of having vitamin D deficiency among nightshift workers (PR = 1.65; 95% CI: 1.23–2.22; *p* = 0.001). Similar findings were observed for vitamin D deficiency/insufficiency (<30 ng/mL). In the unadjusted analysis (Model I), there was a 22% higher probability of having deficiency/insufficiency among nightshift workers (PR = 1.22; 95% CI: 1.08–1.39; *p* = 0.001), and after adjustment, a 17% higher probability of having vitamin D deficiency/insufficiency was observed among nightshift workers (PR = 1.17; 95% CI: 1.02–1.35; *p* = 0.030) compared with dayshift workers.

Complementary analyses (not presented in the tables) were performed considering the following three work shifts: morning/administrative (06:00 AM ≤ hh:hh < 07:00 PM), afternoon (02:00 PM ≤ hh:hh < 10:00 PM), and night (10:00 PM ≤ hh:hh < 06:00 AM). In the bivariate analysis, the “afternoon” shift was associated with higher serum vitamin D levels (median: 24.8 ng/mL, IQR: 20.7–31.6), compared to the “morning/administrative” shift (median: 22.8 ng/mL, IQR: 16.8–29.5) and the “night” shift (median: 18.1 ng/mL, IQR: 13.6–24.2). Additionally, lower prevalence rates of vitamin D deficiency were associated with the “afternoon” shift based on both definition criteria (<20 ng/mL [morning/administrative: 40.8%; afternoon: 16.5%; night: 60%, *p* < 0.001]; <30 ng/mL [morning/administrative: 75.4%; afternoon: 69.4%; night: 90%, *p* = 0.043]). In the multivariable analysis, considering vitamin D deficiency (<20 ng/mL), afternoon-shift workers had a 59% lower probability of having vitamin D deficiency in the fully adjusted model (Model III) (PR = 0.41; 95% CI: 0.24–0.69; *p* = 0.001).

## 4. Discussion

We explored the association between shift work and vitamin D levels in a sample of female workers from a large industrial group located in southern Brazil. A higher probability of having vitamin D deficiency/insufficiency was observed among nightshift workers compared with dayshift workers.

The prevalence rates of vitamin D deficiency and deficiency/insufficiency in our study were 36.5% and 75.7%, respectively, higher than those reported in the review by Pereira-Santos et al. [7] (28.16% and 45.6%, respectively). In this meta-analysis, the southern and southeastern regions of the country showed the highest prevalence rates for deficiency (~30%) and deficiency/insufficiency (~50%). Another study conducted in Brazil using data from three cities in different regions—Northeast (Salvador), Southeast (São Paulo), and South (Curitiba)—reported prevalence rates of 15.3% and 50.9%, respectively [24].

In our study, the participants were fixed-shift workers with a six-day workweek in an indoor environment (a higher-risk setting compared with other occupations), which may explain the higher prevalence of vitamin D deficiency compared with the general population. When comparing our prevalence rates with those reported in previous studies conducted among shift workers, we found similar values. Menezes-Júnior et al. [25] reported a 30.5% prevalence of vitamin D deficiency among rotating shift drivers in southeast Brazil. A similar rate (29%) was observed among male rotating shift workers in an iron ore extraction company [26]. Other studies have revealed even higher prevalence rates: 57.4% among medical residents at a hospital in southern Brazil [27] and 50% among hospital workers in Rome [19].

Higher prevalence rates of vitamin D deficiency among nightshift workers have been reported in previous studies [18,19,20]. This result may be explained mainly by the reduced exposure to sunlight among nightshift workers compared with those working other shifts, owing to various factors involving routine and schedule changes caused by occupational activity [15,16,18]. Vitamin D is primarily produced through cutaneous synthesis from sunlight exposure [1,13]. According to Grigalavicius et al. [28], the most effective way to produce vitamin D, with minimal skin cancer risk, is through sun exposure without sunburn at times closest to midday. Brazil has a generous supply of ultraviolet radiation throughout the year, owing to its continental dimensions. However, it is important to highlight that the southern region has a more pronounced seasonal climate, resulting in significant variability in solar radiation levels between winter and summer, unlike the northern and northeastern regions [29]. Despite differences in latitude and climate, the highest radiation peak, with minimal atmospheric interference, occurs between 10:00 AM and 2:00 PM in any region of the country [30]. In our study, the nightshift workers began their shifts at 10:00 PM and ended them at 6:00 AM, meaning that their commute occurred during periods of low sunlight. In contrast, women working morning and, especially, afternoon shifts were exposed to higher solar radiation peaks during their commutes to and from work, which may explain the difference in prevalence between shifts.

Another hypothesis to explain the higher prevalence of vitamin D deficiency among nightshift workers is related to sleep patterns. It is known that night work opposes the regular circadian rhythm, requiring workers to maintain a sleep–wake cycle misaligned with their natural physiology. Due to this misalignment, nightshift workers are more likely than dayshift workers to report sleep disorders such as insomnia, poor sleep quality, and insufficient sleep duration [31,32]. Circadian disruption and sleep problems affect well-being because of increased daytime sleepiness, fatigue, and the frequent need to use days off to recover from sleep deprivation, and these factors may contribute to reduced social activity and sunlight exposure, leading to increased vitamin D deficiency. Similarly, night work is associated with the development of chronic health conditions, as well as physical inactivity and the interruption of family and social activities [33,34,35,36,37]. This alteration in social patterns owing to health issues may contribute to reduced sun exposure and, consequently, lower blood vitamin D levels.

To further explore the relationship between shift work and vitamin D deficiency, we conducted complementary analyses considering three work shifts (morning/administrative, afternoon, and night). The results showed that female afternoon-shift workers had higher serum levels and lower prevalence rates of vitamin D deficiency and deficiency/insufficiency. This finding has not yet been reported in the scientific literature. However, a possible explanation for this result may be that afternoon-shift workers have lifestyle habits that favor greater sun exposure, such as commuting to work during peak solar radiation hours or engaging in more outdoor activities during the morning hours.

Among the strengths of our study, it is considered one of the first to explore the association between shift work and vitamin D levels in Brazilian female workers. It is noteworthy that this study included a sample of women from a large industrial group located in southern Brazil, incorporating various confounding factors and occupational data obtained through standardized and in-person interviews. Our study also utilized standardized and certified methods and techniques for the collection, storage, and analysis/measurement of serum vitamin D levels. However, this study has some limitations that should be considered. Although our results suggest an association between nightshift work and vitamin D deficiency, there were no questions during the interview about average sun exposure time, type of clothing, the use of sunscreens, or the use of vitamin D supplements. Additionally, data on the timing of physical activity or outdoor activities were not collected for adjustment in the analysis, which could indicate that not only the activity itself, but also the timing and location of the activity, may influence serum vitamin D levels. Although adjustments were performed for confounding factors based on the conceptual determination model [23], the possibility of residual confounding factors due to other lifestyle variables not measured in this study cannot be ruled out. The cross-sectional design of this study, in which exposure and outcomes were measured simultaneously, could not be used to determine causality, meaning that it is impossible to distinguish whether the exposure truly preceded the outcome development. Finally, our study, conducted among female workers on fixed shifts with a six-day workweek in a factory/indoor environment, may not be representative of other shift workers or other population groups owing to the specific characteristics of our sample. Similarly, our population resides in a region with pronounced climate variability, which may not reflect the reality of other regions in Brazil. Therefore, caution should be exercised when generalizing the results to shift workers from other regions, such as the north and northeast of the country.

## 5. Conclusions

The findings of this study indicate a high prevalence of vitamin D deficiency among female shift workers in a large industrial group located in southern Brazil, as well as a significant association between shift work and vitamin D deficiency. Higher probabilities of vitamin D deficiency and deficiency/insufficiency were observed among nightshift workers than among dayshift workers. These results identify this population group as being at risk, highlighting the need for increased attention and health monitoring as well as the development of programs aimed at preventing hypovitaminosis D. It is crucial to highlight that, in future studies on vitamin D, incorporating questions into research tools about supplementation use and sun exposure is vital to achieve more accurate result interpretation and thus advance the field of research.

## Figures and Tables

**Table 1 nutrients-17-01201-t001:** General characteristics of the sample, distribution according to work shift, and serum vitamin D levels (median; IQR) according to demographic, socioeconomic, behavioral, and occupational characteristics among female workers in southern Brazil, 2022 (N = 304).

Characteristics		Vitamin D		Day Shift	Night Shift	
	n (%)	Median (IQR)	*p*-Value	n (%)	n (%)	*p*-Value ^d^
Total		22.9 (17.4–29.1)		264 (86.8)	40 (13.2)	
Demographics						
Age (years)			0.446 ^b^			0.173
18–30	106 (34.9)	21.1 (17.8–28.1)		96 (36.2)	10 (25.0)	
31–40	106 (34.9)	23.7 (17.4–31.6)		93 (35.1)	13 (32.5)	
≥41	92 (30.2)	23.9 (17.5–28.8)		75 (28.7)	17 (42.5)	
Skin color			0.006 ^a^			0.278
White	212 (69.7)	23.6 (18.4–30.8)		187 (70.8)	25 (62.5)	
Other	92 (30.3)	20.1 (14.7–27.9)		77 (29.2)	15 (37.5)	
Marital status			0.708 ^a^			0.379
Without a partner	157 (51.6)	22.8 (17.8–28.8)		139 (52.6)	18 (45.0)	
With a partner	147 (48.4)	23.0 (17.4–29.4)		125 (47.4)	22 (55.0)	
Socioeconomics						
Education level (years)			0.899 ^b^			0.004
≤8	22 (7.2)	25.8 (14.9–31.4)		20 (7.6)	2 (5.0)	
9–11	162 (53.3)	22.7 (17.8–28.1)		131 (49.6)	31 (77.5)	
≥12	120 (39.5)	22.7 (17.7–29.1)		113 (42.8)	7 (17.5)	
Per capita household income (MW) (n = 303) ^c^			0.645 ^b^			0.290
<1	111 (36.6)	22.2 (17.4–30.7)		93 (35.2)	19 (47.5)	
1–2	127 (41.9)	22.9 (16.7–29.1)		114 (43.2)	13 (32.5)	
>2	65 (21.5)	24.0 (18.2–30.8)		57 (21.6)	8 (20.0)	
Head of household			0.461 ^a^			0.455
No	198 (65.1)	22.7 (17.4–28.6)		174 (65.9)	24 (60.0)	
Yes	106 (34.9)	23.4 (17.4–30.9)		90 (34.1)	16 (40.0)	
Behavioral						
Leisure-time physical activity			0.151 ^a^			0.074
No	214 (70.4)	22.5 (17.4–28.3)		181 (68.6)	33 (82.5)	
Yes	90 (29.6)	23.6 (18.7–32.0)		83 (31.4)	7 (17.5)	
Current smoking status			0.962 ^b^			0.265
Non-smoker	232 (76.3)	22.8 (17.6–30.1)		203 (76.9)	29 (72.5)	
Former smoker	53 (17.4)	24.0 (16.8–28.8)		43 (16.3)	10 (25.0)	
Smoker	19 (6.3)	21.2 (17.4–28.3)		18 (6.8)	1 (2.5)	
Alcohol consumption			0.137 ^a^			0.100
No consumption	209 (68.7)	22.2 (17.4–28.3)		177 (67.1)	32 (80.0)	
Consumption of at least once per week	95 (31.2)	23.7 (17.9–32.0)		87 (32.9)	8 (20.0)	
Fruit/vegetable consumption			0.871 ^a^			0.463
No	70 (23.0)	22.1 (17.4–30.8)		59 (22.3)	11 (27.5)	
Yes	234 (77.0)	23.0 (17.4–28.6)		205 (77.7)	29 (72.5)	
Usual sleep duration (hours per day)			0.095			0.032
>6	162 (53.3)	23.5 (18.3–31.4)		147 (55.7)	15 (37.5)	
≤6	142 (46.7)	22.2 (16.6–27.9)		117 (44.3)	25 (62.5)	
Number of pregnancies			0.923 ^b^			0.260
0	112 (36.8)	21.7 (17.6–30.8)		101 (28.3)	11 (27.5)	
1	88 (28.9)	23.7 (17.6–27.5)		77 (29.2)	11 (27.5)	
≥2	104 (34.2)	23.3 (17.3–30.9)		86 (32.6)	18 (45.0)	
Nutritional status (BMI)			0.596 ^b^			0.005
Normal weight	95 (31.2)	22.2 (17.4–30.7)		86 (32.6)	9 (22.5)	
Overweight	116 (38.2)	23.0 (16.8–27.4)		106 (40.1)	10 (25.0)	
Obesity	93 (30.6)	23.6 (18.2–31.4)		72 (27.3)	21 (52.5)	
Occupational						
Length of employment (months)			0.022 ^b^			0.329
≤12	64 (21.0)	24.7 (19.6–32.1)		59 (22.4)	5 (12.5)	
13–36	90 (29.6)	21.1 (17.1–26.9)		78 (29.5)	12 (30.0)	
>36	150 (49.4)	22.7 (16.0–29.4)		127 (48.1)	23 (57.5)	
Job role			0.050 ^a^			<0.001
Factory work (production)	232 (76.3)	22.1 (17.0–28.1)		192 (72.7)	40 (100)	
Administrative	72 (23.7)	24.2 (18.9–31.6)		72 (27.3)	0 (0.0)	
Work shifts			<0.001 ^a^			
Day shift	264 (86.8)	23.4 (18.2–30.8)		-	-	
Night shift	40 (13.2)	18.1 (13.6–24.2)		-	-	

IQR: Interquartile Range; SM: Minimum Wages; Alcohol consumption: refers to the past year; Fruit/vegetable consumption: refers to the past week. ^a^ Mann–Whitney test. ^b^ Kruskal–Wallis test. ^c^ N differs due to missing information (absent). ^d^ Pearson’s chi-square test for proportion heterogeneity.

**Table 2 nutrients-17-01201-t002:** Prevalence of vitamin D deficiency/insufficiency according to demographic, socioeconomic, behavioral, and occupational characteristics among female workers in southern Brazil, 2022 (n = 304).

	Vitamin D			
Characteristics	Deficiency ^a^ (<20 ng/mL)		Deficiency/Insufficiency ^b^ (<30 ng/mL)	
	n (%)	*p*-Value ^c^	n (%)	*p*-Value ^c^
Total	111 (36.5)		230 (75.7)	
Demographics				
Age (years)		0.678		0.326
18–30	42 (39.6)		84 (79.2)	
31–40	38 (35.8)		75 (70.7)	
≥41	31 (33.7)		71 (77.2)	
Skin color		0.003		0.201
White	66 (31.1)		156 (73.6)	
Other	45 (48.9)		74 (80.4)	
Marital status		0.302		0.954
Without a partner	53 (33.8)		119 (75.8)	
With a partner	58 (39.5)		111 (75.5)	
Socioeconomics				
Education level (years)		0.632		0.382
≤8	8 (36.4)		14 (63.6)	
9–11	63 (38.9)		125 (77.2)	
≥12	40 (33.3)		91 (75.8)	
Per capita household income (MW) (n = 303) ^c^		0.973		0.853
<1	41 (36.9)		83 (74.8)	
1–2	47 (37.0)		98 (77.2)	
>2	23 (35.4)		48 (73.8)	
Head of household		0.941		0.538
No	72 (36.4)		152 (76.8)	
Yes	39 (36.8)		78 (73.6)	
Behavioral				
Leisure-time physical activity		0.314		0.136
No	82 (38.3)		167 (78.0)	
Yes	29 (32.2)		63 (70.0)	
Current smoking status		0.800		0.883
Non-smoker	87 (37.5)		174 (75.0)	
Former smoker	18 (34.0)		41 (77.4)	
Smoker	6 (31.6)		15 (78.9)	
Alcohol consumption		0.343		0.090
No consumption	80 (38.3)		164 (78.5)	
Consumption of at least once per week	31 (32.6)		66 (69.5)	
Fruit/vegetable consumption		0.901		0.209
No	26 (37.1)		49 (70.0)	
Yes	85 (36.3)		181 (77.3)	
Usual sleep duration (hours per day)		0.052		0.079
>6	51 (31.5)		116 (71.6)	
≤6	60 (42.2)		114 (80.3)	
Number of pregnancies		0.948		0.263
0	41 (36.6)		83 (74.1)	
1	31 (35.2)		72 (81.8)	
≥2	39 (37.5)		75 (72.1)	
Nutritional status (BMI)		0.846		0.170
Normal weight	36 (37.9)		71 (74.7)	
Overweight	40 (34.5)		94 (81.0)	
Obesity	35 (37.6)		65 (69.9)	
Occupational				
Length of employment (months)		0.157		0.138
≤12	17 (26.6)		43 (67.2)	
13–36	37 (41.1)		73 (81.1)	
>36	57 (38.0)		114 (76.0)	
Job role		0.229		0.160
Factory work (production)	89 (38.4)		180 (77.6)	
Administrative	22 (30.6)		50 (69.4)	
Work shifts		0.001		0.023
Day shift	87 (32.9)		194 (73.5)	
Night shift	24 (60.0)		36 (90.0)	

MW: Minimum Wages; Alcohol consumption—refers to the past year; Fruit/vegetable consumption—refers to the past week. ^a^ Vitamin D deficiency according to the Institute of Medicine [1]. ^b^ Endocrine Society [13]. ^c^ Pearson’s chi-square test for proportion heterogeneity.

**Table 3 nutrients-17-01201-t003:** Unadjusted and adjusted prevalence ratios (PRs) and their respective 95% confidence intervals (95% CI) for the association between shift work and vitamin D deficiency (<20 ng/mL) and deficiency/insufficiency (<30 ng/mL) among female workers in southern Brazil, 2022 (N = 304).

Work Shifts	Vitamin D Deficiency (<20 ng/mL)	Model I	Model II ^b^	Model III ^c^
	n (%)	PR (95% CI)	PR (95% CI)	PR (95% CI)
Day shift	87 (32.9)	1.00	1.00	1.00
Night shift	24 (60.0)	1.82 (1.34–2.47)	1.68 (1.25–2.27)	1.65 (1.23–2.22)
*p*-value ^a^		<0.001	0.001	0.001
	Vitamin D deficiency/insufficiency (<30 ng/mL)	Model I	Model II ^d^	Model III ^e^
Work shifts	n (%)	PR (95% CI)	PR (95% CI)	PR (95% CI)
Day shift	194 (73.5)	1.00	1.00	1.00
Night shift	36 (90.0)	1.22 (1.08–1.39)	1.19 (1.04–1.363)	1.17 (1.02–1.35)
*p*-value ^a^		0.001	0.010	0.030

Model I: unadjusted analysis. ^a^ *p*-value for Wald test for proportion heterogeneity obtained through Poisson regression with robust variance. ^b^ Model II: adjusted analysis for demographic (skin color) and behavioral (sleep hours) variables. ^c^ Model III: adjusted analysis for Model II + occupational variables (length of employment). ^d^ Model II: adjusted analysis for demographic variables (skin color) and behavioral variables (physical activity, alcohol consumption, sleep duration, and nutritional status). ^e^ Model III: adjusted analysis for Model II + occupational variables (length of employment and job role).

## Data Availability

The raw data supporting the conclusions of this article will be made available by the authors on request. The original contributions presented in the study are included in the article, further inquiries can be directed to the corresponding author.

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
