# Peer review of "Association Between Shift Work and Vitamin D Levels in Brazilian Female Workers"

_nutrients, 2025, doi:10.3390/nu17071201_

Round 1
Reviewer 1 Report
Comments and Suggestions for Authors
The authors conducted an interesting study aimed at establishing the relationship between shift work and vitamin D levels among female workers. They formulated a research hypothesis that night shift workers have a higher incidence of vitamin D deficiency.
To verify this hypothesis, the authors conducted a cross-sectional study among a group of 304 workers, including production sector workers engaged in fixed shifts with a six-day workweek schedule and 72 administrative sector workers. in the metropolitan region of Porto Alegre, Rio Grande do Sul, Brazil. The study consisted of conducting an interview according to the author's questionnaire, as well as taking a blood sample and determining serum levels of 25(OH)D. During the interview, basic data on demographic, socioeconomic, behavioral, and occupational characteristics of the study participants were obtained. Behavioral characteristics investigated included leisure-time physical activity, assessed based on the report of practicing any physical activity for leisure, sports, or exercise in the past week, excluding commuting. Occupational characteristics allowed us to distinguish morning, afternoon, and night shift workers. Valid, reliable methods were used to measure serum levels of 25(OH)D. However, the survey questionnaire prepared for a larger research project entitled 84 “Health Conditions of Shift-Working Women: Longitudinal Occupational Health Study 85 (ELO Saúde)” lacked several simple, detailed questions about sun exposure, which became a significant obstacle to substantiating the suggestion that this exposure is the most important factor influencing vitamin D levels in people working at different times of the day.
As a result of the conducted studies and analyses, a higher probability of vitamin D deficiency was observed among night-shift workers in comparison to day workers, which the authors explain by the specific time of commuting to and returning from work during periods of weak sunlight. A more detailed analysis taking into account three work shifts (morning/administrative, afternoon, and night) showed that female afternoon-shift workers had higher serum levels and lower prevalence rates of vitamin D deficiency and deficiency/insufficiency. The authors conclude that a possible explanation for this result may be that afternoon-shift workers have lifestyle habits that favor greater sun exposure, such as commuting to work during peak solar radiation hours or engaging in more outdoor activities during the morning hours. However, the importance of being and being active outdoors during the period of the greatest sunlight was not well documented in this study. During the interview, there were no questions about average sun exposure time, type of clothing, and use of sunscreens, and no questions about use of vitamin D supplements. The authors listed these factors among the risk factors for vitamin D deficiency that they knew, but did not include them in the interviews. Despite these shortcomings, which cannot be improved, I believe that the article contains so much important information that it deserves to be published. However, I believe that the authors should formulate a note addressed to other researchers indicating the need to exercise special attention at the stage of constructing research tools, because the lack of several simple questions in the questionnaire may prevent the correct interpretation of the obtained results.
Author Response
09-March-2025
Dear Reviewer,
Thank you very much for taking the time to review this manuscript. We carefully considered your comment. Here, we explain how we revised the paper and answer your suggestion. The changes were highlighted in the "Main Document" using a red font color.
All authors reviewed the changes and approved the final version of the manuscript.
Comment: ‘I believe that the authors should formulate a note addressed to other researchers indicating the need to exercise special attention at the stage of constructing research tools, because the lack of several simple questions in the questionnaire may prevent the correct interpretation of the obtained results’.
We revised and added this sentences in the ‘Discussion’ and “Conclusion” sections of the manuscript to address the suggestion:
In the Discussion: Although our results suggest an association between night-shift work and vitamin D deficiency, during the interview there were no questions about average sun exposure time, type of clothing, and use of sunscreens, and no questions about use of vitamin D supplements.
In the Conclusion: It is crucial to highlight that, in future studies on vitamin D, incorporating questions about supplementation use and sun exposure into research tools is vital for achieving more accurate result interpretation and advancing the field of research.
Thank you for your consideration.
Sincerely,
Ingrid Stähler ohl
Anderson Garcez
Janaína Cristina da Silva
Harrison Canabarro de Arruda
Raquel Canuto
Vera Maria Paniz Vieira
Maria Teresa Anselmo Olinto
Reviewer 2 Report
Comments and Suggestions for Authors
The manuscript “ Association between shift work and vitamin D levels in Brazil-2 ian female workers ”by Ingrid Stähler Kohl et al aimed to evaluate the association between shift work and vitamin D levels among female workers.; the conclusion was that the probability of vitamin D deficiency or insufficiency was higher among night-shift workers compared to day-shift workers.
COMMENTS
1).. The authors need to better clarify the characteristics of women who work night shifts. How many night shifts do they work each week? Do they always work at night or is there a periodic rotation?
2).. The reduced vitamin D levels in women who work night shifts could be influenced by differences between them and those who work day shifts in terms of job type (administrative or manual), skin color, socioeconomic status, and other factors. Therefore, it would be useful to include a table comparing the two groups (night shift and day shift) based on age, skin color, education and other relevant characteristics.
3). The authors should indicate more precisely in which period of the year they carried out the vitamin D level assessments.
4). Other limitations of the study are represented by the lack of information on the number of pregnancies, body weight and intake of vitamin D supplements
Author Response
09-March-2025
Dear Reviewer,
Thank you very much for taking the time to review this manuscript. We carefully considered the comments. Please find below the detailed responses and the corresponding revisions/corrections. Here, we explain how we revised the paper and address the reviewers' comments item-by-item. The changes were highlighted in the "Main Document" using a red font color.
All authors reviewed the changes and approved the final version of the manuscript.
Comments 1: How many night shifts do they work each week? Do they always work at night or is there a periodic rotation?
We revised and added this sentence in the ‘Materials and Methods’ sections of the manuscript to make it clearer: ‘The permanent production line shift workers, who work a total of 44 hours per week, follow a schedule of six consecutive fixed shifts (Monday through Saturday), followed by one day off , and are distributed across the following fixed shifts:’.
Comments 2: It would be useful to include a table comparing the two groups (night shift and day shift) based on age, skin color, education and other relevant characteristics.
We appreciate your suggestion. To address it, we included in Table 1 a description according to the work shift of the participants in relation to all variables studied in the paper. This way, it is possible to better visualize and identify the characteristics of the sample. Additionally, we included in the ‘Results’ section the characteristics of the sample according to the work shift: ‘Regarding work shifts, all night-shift workers were production workers. The vast majority (77.5%) had 9 to 11 years of education, while only 17.5% had 12 or more years of education. Additionally, 82.5% did not engage in leisure-time physical activity, 62.2% slept 6 hours or less per day, and 52.5% were obese’.
Comments 3: The authors should indicate more precisely in which period of the year they carried out the vitamin D level assessments.
We revised and added this sentence in the ‘Materials and Methods’: ‘The laboratory tests were conducted during the same period, covering the winter, spring, and summer seasons in Brazil’.
Comments 4: Other limitations of the study are represented by the lack of information on the number of pregnancies, body weight and intake of vitamin D supplements
We revised and included in Table 1 the variables of number of pregnancies and nutritional status for better representation of the sample. Additionally, we added a sentence in the ‘Discussion’ section to address the limitations regarding the lack of information on supplementation: ‘Although our results suggest an association between night-shift work and vitamin D deficiency, during the interview there were no questions about average sun exposure time, type of clothing, and use of sunscreens, and no questions about use of vitamin D supplements’.
Thank you for your consideration.
Sincerely,
Ingrid Stähler ohl
Anderson Garcez
Janaína Cristina da Silva
Harrison Canabarro de Arruda
Raquel Canuto
Vera Maria Paniz Vieira
Maria Teresa Anselmo Olinto
Round 2
Reviewer 2 Report
Comments and Suggestions for Authors
The Authors adequately responded to to the comments of the referee.